

# Identification, expression and variation of the *GNPDA2* gene, and its association with body weight and fatness traits in chicken

Hongjia Ouyang, Huan Zhang, Weimin Li, Sisi Liang, Endashaw Jebessa, Bahareldin A. Abdalla and Qinghua Nie

Department of Animal Genetics, Breeding and Reproduction, College of Animal Science, South China Agricultural University, Guangzhou, China
Guangdong Provincial Key Lab of Agro-Animal Genomics and Molecular Breeding, Guangzhou, China

Corresponding author
Qinghua Nie, nqinghua@scau.edu.cn

## ABSTRACT

**Background.** The *GNPDA2* (glucosamine-6-phosphate deaminase 2) gene is a member of Glucosamine-6-phosphate (GlcN6P) deaminase subfamily, which encoded an allosteric enzyme of GlcN6P. Genome-wide association studies (GWAS) have shown that variations of human *GNPDA2* are associated with body mass index and obesity risk, but its function and metabolic implications remain to be elucidated. The object of this study was to characterize the gene structure, expression, and biological functions of *GNPDA2* in chickens.

**Methods.** Variant transcripts of chicken *GNPDA2* and their expression were investigated using rapid amplification of cDNA ends (RACE) system and real-time quantitative PCR technology. We detected the *GNPDA2* expression in hypothalamic, adipose, and liver tissue of Xinghua chickens with fasting and high-glucose-fat diet treatments, and performed association analysis of variations of *GNPDA2* with productive traits in chicken. The function of *GNPDA2* was further studied by overexpression and small interfering RNA (siRNA) methods in chicken preadipocytes.

**Results.** Four chicken *GNPDA2* transcripts (*cGNPDA2*-a~*cGNPDA2*-d) were identified in this study. The complete transcript *GNPDA2*-a was predominantly expressed in adipose tissue (subcutaneous fat and abdominal fat), hypothalamus, and duodenum. In fasting chickens, the mRNA level of *GNPDA2* was decreased by 58.8% ($P < 0.05$) in hypothalamus, and returned to normal level after refeeding. Chicken fed a high-glucose-fat diet increased *GNPDA2* gene expression about 2-fold higher in adipose tissue ($P < 0.05$) than that in the control (fed a basal diet), but decreased its expression in hypothalamus. Two single-nucleotide polymorphisms of the *GNPDA2* gene were significantly associated with body weight and a number of fatness traits in chicken ($P < 0.05$).

**Conclusion.** Our findings indicated that the *GNPDA2* gene has a potential role in the regulation of body weight, fat and energy metabolism in chickens.

## INTRODUCTION

Obesity is a common nutritional disorder and increase risk of several diseases, such as hypertension, cardiovascular disease and type II diabetes (*Kopelman*, *2000*; *Flegal et al.*, *2007*; *Finkelstein et al.*, *2008*). It becomes a major public health problem because it has dramatically increased worldwide during the past years. Obesity is modulated by environmental and genetic factors. The modern environment contributes to the increasing prevalence of obesity, but genetic factors modulate the susceptibility of each individual (*Maes, Neale & Eaves*, *1997*; *Barsh, Farooqi & O'Rahilly*, *2000*).

The *GNPDA2* gene encodes an allosteric enzyme of Glucosamine-6-phosphate deaminase (GlcN6P), which can catalyzes the reversible conversion of D-glucosamine-6-phosphate into D-fructose-6-phosphate and ammonium (*Arreola et al.*, *2003*). The GlcN6P enzyme was initially characterized in pig kidney; it has been also identified in a variety of organisms (*Leloir & Cardini*, *1956*). This enzyme was annotated include hydrolase activity and glucosamine-6-phosphate deaminase activity, and related the pathways of metabolism and amino sugar and nucleotide sugar metabolism. The SNP near the human *GNPDA2* gene (rs10938397) was first identified to be associated with both body mass index (BMI) and weight in adult of European by GWAS in 2009 (*Willer et al.*, *2009*), and then it was confirmed in adult of various populations, including Europeans, Americans, East Asians and Arabians (*Hotta et al.*, *2009*; *Speliotes et al.*, *2010*; *Wen et al.*, *2012*; *Gong et al.*, *2013*; *Tomei et al.*, *2015*). The variations of *GNPDA2* were also found to be associated with BMI or risk of obesity in children and old age population (*Elks et al.*, *2010*; *Mejía-Benítez et al.*, *2013*; *Murphy et al.*, *2013*; *Xi et al.*, *2013a*; *Pillay et al.*, *2015*). Furthermore, it also modulated the susceptibility of hypertension and type II diabetes (*Takeuchi et al.*, *2011*; *Robiou-du-Pont et al.*, *2013*; *Xi et al.*, *2013b*; *Xi et al.*, *2014*; *Kong et al.*, *2015*).

Fatness traits are important economic traits in chickens because of its relation to meat quality of broiler, suitable content of fat can improve the meat quality (tenderness, juiciness and flavor), but excess fat deposition can greatly reduce the meat quality (*Castellini, Mugnai & Bosco*, *2002*; *Zerehdaran et al.*, *2004*; *Petracci et al.*, *2013*). In addition, large amounts of fat deposition also can reduce the feed efficiency, even increasing the risk of disease in chicken (*Boekholt et al.*, *1994*). In humans, *GNPDA2* has been found to be associated with BMI and fat deposition, but the characteristics and functions of *GNPDA2* in chickens have not been reported. In this study, we hypothesize that *GNPDA2* is also involved in body weight gain and fat mass accumulation in chickens. Therefore, we identified the chicken *GNPDA2* gene and monitored its mRNA level in various tissues under different conditions of nutrition, and performed association analysis of variations of *GNPDA2* with fatness traits in chickens, so as to characterize the functions of the *GNPDA2* gene in the fat metabolism of chickens.

## MATERIALS AND METHODS

### Animals and DNA samples

A total of 45 Xinghua chickens (20 chickens (10 males and 10 females) at 14 weeks of age; 25 female chickens at 23 weeks of age) were obtained from Poultry Farm of the South China Agricultural University (SCAU). All birds were raised in individual cages, kept in

identical light/dark cycles and had ad libitum access to water throughout the experimental period. Birds were allocated to the one of three different treatments (T) as follows: T1, the 14-week-old chickens were divided into two groups (in each group; $n = 10$, 5 males and 5 females) and fed a high-glucose-fat diet (Table S1) or a basal diet for 4 weeks, respectively. T2, the 23-week-old chickens with significant different body weight were divided into high body weight group ($n = 5$, $1388.5 \pm 122.4$ g) and low body weight group ($n = 5$, $1132.7 \pm 108.5$ g). In T3, the remain 15 23-week-old female chickens were divided into three groups of five birds each and subjected to the following treatments: (I) Fed a basal diet ad libitum (Control); (II) Fasted for two days (Fasted); (III) Fasted for two days and refed a basal diet for one day (Re-Fed).

The DNA samples from an $F_2$ resource population which derived from reciprocal crossed between Xinghua (XH) and White Recessive Rock (WRR) chickens were used for *GNPDA2* variations identifying and association analysis (*Lei et al.*, *2005*). The population was made up of 17 full-sibling families, which had 434 F2 individuals (221 male and 213 female chickens) with a detailed record of growth traits, carcass traits, and meat quality traits.

## Primers and siRNA

All primers were designed using Premier Primer 5.0 software (Premier Bio-soft International, Palo Alto, CA, USA) and synthesized by Biosune Co. Ltd (Shanghai, China). Primers G1 and G2 were used to clone c*GNPDA2*. G5′-outer and G5′-inner were used for 5'RACE PCR, G3′-outer and G3′-inner were used for 3'RACE PCR. G3 and G4 were used for real-time-PCR analysis of chicken *GNPDA2* and $\beta$-actin, respectively. Primers G5–G11 were used to identify and genotype SNPs of c*GNPDA2*. All SR1~SR3 siRNAs were synthesized by GenePharma Co. Ltd (Suzhou, China). Scrambled siRNA was also synthesized as negative control. Primers of *GNPDA2* are summarized in Table S2. Primers for real time quantitative PCR (RT-qPCR) of other *GNPDA2*-related genes are summarized in Table S3.

## RNA isolation, cDNA synthesis and real-time PCR analysis

Upon termination of the experiment, chickens were euthanized and 16 tissues (cerebrum, cerebellum, hypothalamus, pituitary, abdominal fat, subcutaneous fat, breast muscle, heart, liver, spleen, lungs, kidney, muscular stomach, glandular stomach, duodenum and ovary/testis) were rapidly collected, immediately frozen in liquid nitrogen and stored at $-80\,°C$ prior to use. Total RNA was isolated using Trizol reagent (Invitrogen, Foster City, CA, USA), following the recommended manufacturer's protocol. The quality and quantity of RNA samples were detected by 1.5% agarose gel electrophoresis and based on absorbance OD (optical density) at 260/280 nm ratio, respectively. For the cDNA synthesis, 2 μg total RNA was subjected to reverse transcription with the use of oligo(dT)$_{18}$ as the primer and reverse transcriptase was performed using RevertAid$^{TM}$ First Strand cDNA Synthesis Kit (Fementas, Waltham, MA, USA) in total 20 μL reaction volume. The relative quantity of mRNA was detected using SsoFast Eva Green Supermix (BIO-RAD, Hercules, USA) in a final volume of 20 μL and performing in CFX9600 (BIO-RAD) under the following conditions: $95\,°C$ for 3 min, followed by 40 cycles of 10 s at $95\,°C$, 30 s at annealing temperature ($58–62\,°C$), 30 s at $72\,°C$ and melt curve by $65\,°C$ to $95\,°C$,

increment 0.5 °C for 5 s. Each sample was assayed in triplicate, and chicken $\beta$−actin was used as reference gene. The specificity of product was decided by the solubility curve, and the quantitative values were obtained from the threshold PCR quantification cycle (Cq). The relative mRNA level in each sample was calculated using comparative equation $2^{-\Delta Ct}(\Delta Ct = Ct_{target\ gene} - Ct_{\beta\text{-actin}})$ (PCR efficiency was considered as 100%) by CFX Manager software (version 2.1). Fold-change values were calculated using the comparative $2^{-\Delta\Delta Ct}(\Delta\Delta Ct = \Delta Ct_{target\ sample} - \Delta Ct_{control\ sample})$ method.

## 5'RACE and 3'RACE PCR

The RACE system was performed to obtain the full-length cDNA sequence of chicken *GNPDA2*. The hypothalamus and adipose tissue total RNA were used as template for RACE PCR. RACE PCR was performed with the SMARTer RACE cDNA Amplification Kit (Clontech, Osaka, Japan) following the manufacturer's instructions. Products of RACE PCR were cloned into pMD-18T vector (Takara, Osaka, Japan) according to the manufacturer's protocol, and sequenced by Invitrogen Co. Ltd (Guangzhou, China).

## *GNPDA2* blast analysis and phylogenetic analysis

The *GNPDA2* AA sequences of the other 17 species were obtained from Genebank (Table S4). The obtained cDNA sequences were analyzed by BLAST (https://blast. ncbi.nlm.nih.gov/Blast.cgi). On the basis of the 18 *GNPDA2* sequences, a phylogenetic tree was constructed by using Neighbor-joining method of the MEGA 4.1 software (http://www.megasoftware.net/mega41.html).

## SNP identification and genotyping

Variations of *cGNPDA2* were identified and genotyped in DNA samples of $F_2$ resource population (XH&WRR) using PCR amplification and sequencing. PCR was performed in volume of 50 µL of a mixture containing 50 ng of chicken genomic DNA, 10 pmol of primers and 25 µL PCR Master Mix (Transgen, Beijing, China), and using the following protocol: 94 °C for 3 min, followed by 32 cycles of 30 s at 94 °C, 30 s at annealing temperature (58–62 °C; Table S3), 30 s at 72 °C and a final extension of 72 °C for 5 min. PCR products were sequenced by Invitrogen Co. Ltd (Guangzhou, China). PCR-restriction fragment length polymorphism (PCR-RFLP) was also used to genotyping SNPs. PCR products were digested by fasted restriction enzymes (Fementas) following the manufacturer's instructions, and then detected through 2% agarose gel electrophoresis.

## Plasmid construction, cell culture and transfection

The coding sequences of c*GNPDA2*-a were amplified from chicken abdominal fat cDNA using PCR, and then cloned into pcDNA3.1(+) vector (Invitrogen, Carlsbad, CA, USA) according to the protocols of the manufacturer, using the *XhoI* and *BamHI* (Fementas) restriction sites. Chicken preadipocytes were obtained from 20 days old Xinghua chicks as previous study (*Ramsay & Rosebrough*, *2003*). The cell were maintained in Dulbecco's modified Eagle medium (DMEM)/F-12 (Gibco, Grand Island, NY, USA) supplemented with 10% (v/v) foetal bovine serum (Hyclone, Logan, UT, USA), and 100 µg/mL penicillin/streptomycin (Invitrogen) at 37 °C with 5% $CO_2$, humidified atmosphere.

Preadipocytes were seeded in 12-well plate at a density of $5.0 \times 10^4$ cells/well, and were transfected plasmid (1 µg/per well) or siRNA (50 nM) using Lipofectamine 3000 (Invitrogen) (1 µL/per well) when the cells growth reach 70%~80% confluence. After 48 h, all cells were collected to assay the mRNA level of *GNPDA2* and its related genes. The pcDNA3.1(+) vector without insert and scrambled siRNA were used as an internal control, respectively.

### Statistical analysis

All values are presented as means ± S.E.M. The threshold for significance was set at $P < 0.05$ and for high significance at $P < 0.01$. Association analysis of SNPs with productive traits were performed using the General Linear Models Procedures of SAS 9.0 (SAS Institute Inc., Cary, NC, USA) and the genetic effects were analyzed by a mixed procedure according to the following model (*Lei et al.*, *2005*):

$$Y_{ijkl} = \mu + S_i + G_j + H_k + F_l + G_i * S_j + G_i * H_k + G_i * F_l + e_{ijkl}$$

where $Y$ = the traits phenotypic values; $\mu$ = the overall population mean; $S$ = the fixed effect of gender; $G$ = the fixed effect of genotype; $H$ = the fixed effect of hatch; $F$ = the fixed effect of family; $e$ = the random residuals.

### Animal Ethics

The experimental procedures used in this study met the guidelines of the Animal Care and Use Committee of the South China Agricultural University (SCAU) (Guangzhou, People's Republic of China) and were approved by SCAU with approval number SCAU# 0011. Animal experiments were handled in compliance with the regulations and guidelines established by the Animal Care and Use Committee of SCA, and all efforts were made to minimize animal suffering.

## RESULTS

### The chicken *GNPDA2* cDNA and variant transcripts

Four different *GNPDA2* transcripts were identified in chicken in this study, the length of each of them as follows: c*GNPDA2*-a (NCBI accession number: JX048609) was 1,806 bp, c*GNPDA2*-b (JX048610) was 1,621 bp, c*GNPDA2*-c (KF296359) was 1,522 bp and c*GNPDA2*-d (KF296360) was 1,715 bp. The results of UCSC genome BLAST showed that the chicken *GNPDA2* was 7,453 bp long, which located at chromosome 4 and spanned from 67202451 to 67209903. Here, our result c*GNPDA2*-a, the predominant transcript of *GNPDA2*, comprised seven exons and six introns. Compared with c*GNPDA2*-a, c*GNPDA2*-b was deleted the whole exon5 (185 bp), c*GNPDA2*-c was deleted the exon5 and part of the exon6 (99 bp), and c*GNPDA2*-d has a 91 bp deletion in exon2. Exon 5 and 6 are 'hot spots' for alternative splicing (Fig. 1A). Due to the deletions located at coding region, four *GNPDA2* transcripts have different open reading frame (ORF). The ORF of c*GNPDA2*-a was 828 bp and encoding 275 amino acids (AA). The ORF length of the other three c*GNPDA2* transcripts were 438 bp for c*GNPDA2*-b, 435 bp for c*GNPDA2*-b and 737 bp for c*GNPDA2*-d, respectively (Fig. 1B).

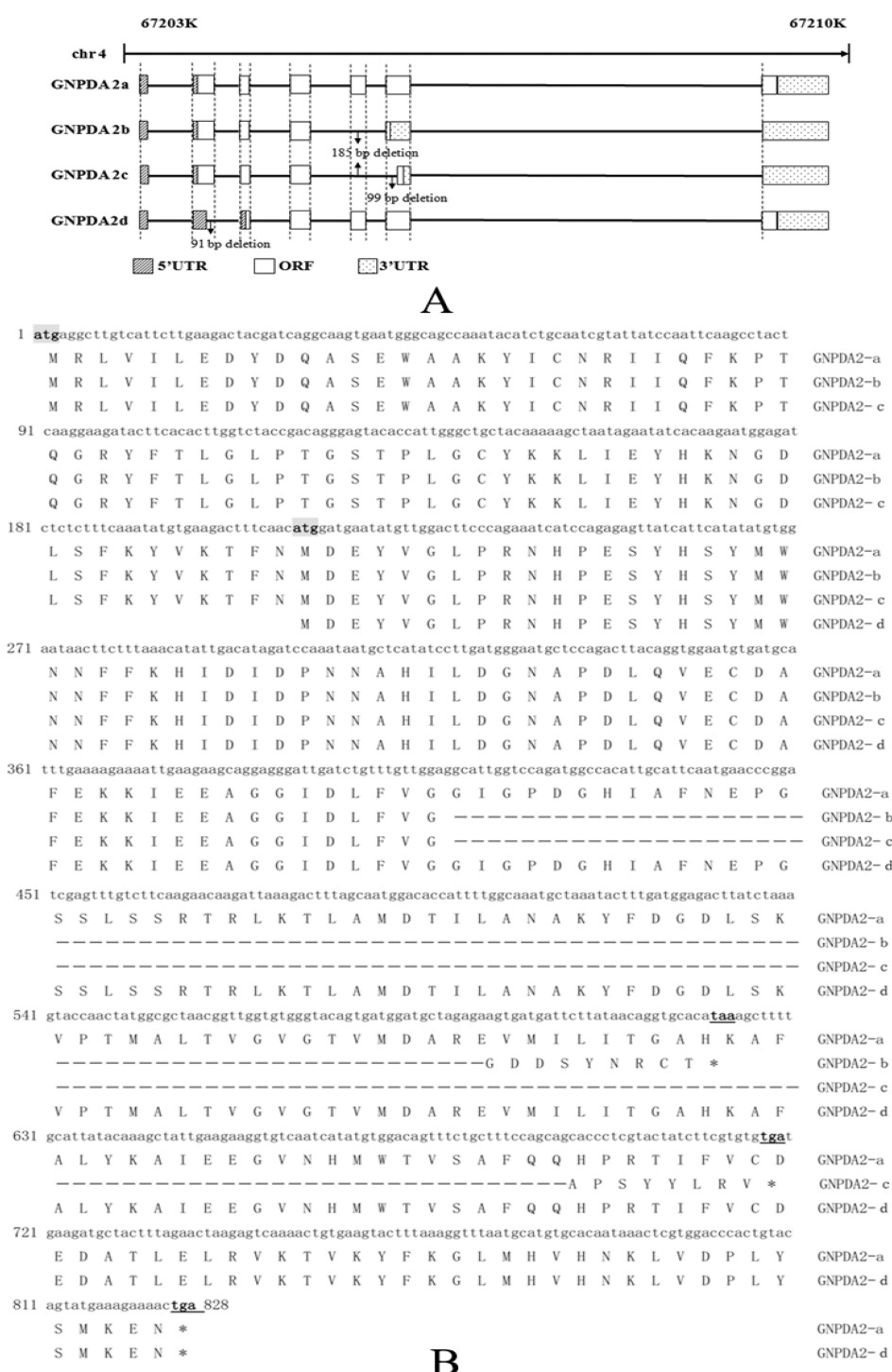

**Figure 1 Characterization of four chicken *GNPDA2* variant transcripts.** (A) Genomic structure of chicken *GNPDA2*. UTR, untranslated region; ORF, open reading frame. (B) The open reading frame of four cGNPDA2 variant transcripts. The letter atg with shadow is initiation codon, and the letter taa or tga with underline is termination codon.

## Sequence alignment and phylogenetic analysis

*GNPDA2* gene is a member of GlcN6P deaminase subfamily. Analysis by using NCBI BLAST (Basic Local Alignment Search Tool) showed that the coding DNA sequences of *GNPDA2* were highly conserved, and has conserved domain of GlcN6P deaminase in chicken and other 17 species. The *GNPDA2* sequence homology of chicken with other birds and mammalians were all above 90%, especially the homology with duck up to 99.3% (Fig. S1A). The phylogenetic tree based on *GNPDA2* homology showed that 18 species were divided into four distinct groups, and they were birds (chicken, turkey, duck, parrot, pigeon, zebra finch and sparrow), mammals (human, cattle, dog, pig, horse, goat, sheep, mouse and rat), frog and zebrafish (Fig. S1B).

## SNPs identification and its association analysis with productive traits

Chicken *GNPDA2* was located at chromosome 4 and spanned in 7,453 bp (NC_006091.3). In $F_2$ resource population (XH&WRR) chicken, a total of 25 SNPs were identified in the full length chicken *GNPDA2* DNA through PCR sequencing (Table 1). Among these 25 SNPs, 6 SNPs were novel ones, but the other 19 SNPs have already been reported in NCBI SNP database (https://www.ncbi.nlm.nih.gov/snp). According to the transcript c*GNPDA2*-a, two SNPs were found in the coding sequences, seven SNPs were located in 3'UTR sequences and the other 16 SNPs were located in intron sequences.

The c*GNPDA2* SNPs in coding sequences and 3'UTR sequences were genotyped in $F_2$ resource population (XH&WRR) chicken through PCR sequencing, and then the results were association analysis with productive traits. We found that the two SNPs (g.6667C > T and g.7393C > T) in 3'UTR (Fig. 2A) were significantly associated with a number of productive traits in chicken (Table 2). The genotyped results of these two SNPs were also confirmed by PCR-RFLP (Fig. 2B). PCR products of g.6667C > T (1,077 bp) and g.7393C > T (648 bp) were digested by *Dde I* and *Ssp I* respectively, and then detected enzyme-digested fragments to genotyed the g.6667C > T (TT, 1,037 bp; TC, 1077/703/374 bp; CC, 703/374 bp) and g.7393C > T (TT, 648 bp; TC, 648/587/61 bp; CC, 587/61 bp). The g.6667C > T and g.7393C > T (rs14486239) were both significantly associated with body weight during early stage (1–28 days old) of growth (Fig. 2C). The g.6667C > T was also significantly associated with chest depth (CD) and abdominal fat pad weight (AFW) (Fig. 2D). The g.7393C > T was significantly associated with small intestine length (SIL), leg muscle weight (LMW), dressed weight (DW), eviscerated weight (EW) and semi-eviscerated weight (SEW) (Fig. 2D).

## Tissue specific expression of *GNPDA2* in XH chicken

Relative *GNPDA2* mRNA expression in different tissues was detected by RT-qPCR and the quantified data are shown in Fig. 3. Chicken *GNPDA2* gene was expressed in all 17 tested tissues, and there were no significantly different ($P > 0.05$) in expression level of all tissues between female and male chickens (Fig. 3). Chicken *GNPDA2* gene expression level was highest in abdominal fat, duodenum and hypothalamus, and obvious mRNA expression in subcutaneous fat liver, lung, spleen and pituitary was also detected. However, the low expression level was observed in other tissue, especially in kidney and muscular
**Table 1  25 SNPs identified in the full length of chicken GNPDA2 DNA sequence.**

| NO. | Site in DNA | SNPs | Region | NCBI number |
| --- | --- | --- | --- | --- |
| 1 | nt 1157 | C→T | Intron 1 | rs313319798 |
| 2 | nt 1192 | G→A | Intron 1 | rs312694820 |
| 3 | nt 1250 | A→T | Intron 1 | – |
| 4 | nt 1315 | T→C | Intron 1 | rs313558025 |
| 5 | nt 1347 | G→A | Intron 1 | – |
| 6 | nt 2119 | T→C | Exon 4 | rs316460349 |
| 7 | nt 2412 | G→A | Exon 4 | rs316364048 |
| 8 | nt 3404 | A→T | Intron6 | rs14486231 |
| 9 | nt 3590 | C→T | Intron 6 | rs312766474 |
| 10 | nt 4231 | -/T | Intron 6 | rs14486233 |
| 11 | nt 5436 | A→G | Intron 6 | rs317780046 |
| 12 | nt 5527 | T→C | Intron 6 | – |
| 13 | nt 5605 | A→G | Intron 6 | rs317672439 |
| 14 | nt 5659 | A→G | Intron 6 | rs312941054 |
| 15 | nt 5841 | C→G | Intron 6 | rs312807283 |
| 16 | nt 6091 | C→G | Intron 6 | rs794508049 |
| 17 | nt 6098 | C→T | Intron 6 | – |
| 18 | nt 6178 | G→A | Intron 6 | rs312926139 |
| 19 | nt 6667 | C→T | 3'UTR | – |
| 20 | nt 6822 | T→A | 3'UTR | – |
| 21 | nt 7090 | A→G | 3'UTR | rs313999192 |
| 22 | nt 7277 | A→G | 3'UTR | rs14486237 |
| 23 | nt 7306 | T→C | 3'UTR | rs315312077 |
| 24 | nt 7226 | C→G | 3'UTR | rs14486238 |
| 25 | nt 7393 | C→T | 3'UTR | rs14486239 |

stomach. In addition, we compared the mRNA level of *GNPDA2* in four tissues (abdominal fat, subcutaneous fat, hypothalamus, and liver) of 23 weeks old female XH chickens with different body weight (high or low body weight). The results showed that *GNPDA2* has higher expression in the hypothalamus of high body weight than that in the low body weight chickens (Fig. 4A).

**The effects of dietary status on *cGNPDA2* mRNA level**

After feeding chickens with high-glucose-fat diet, the mRNA level of *GNPDA2* was increased about 2-fold in abdominal fat ($P < 0.05$) and subcutaneous fat ($P < 0.01$) than that in the control group (basal diet) of both female and male XH chickens. In liver and hypothalamus, the mRNA level of *GNPDA2* was decreased of both female and male XH chickens, especially decreased by 38% ($P < 0.05$) in hypothalamus of female chicken (Fig. 4B). In fasted chickens, the mRNA level of *GNPDA2* was decreased by 58.8% ($P < 0.05$) in hypothalamus, and returned to normal level after refeeding. However, no significant difference ($P > 0.05$) was found in adipose tissue and liver among the three experimental groups (control, fasted and re-fed), but the mRNA expression level in abdominal fat and subcutaneous fat tended to be slightly higher ($P > 0.05$) with fasted group (Fig. 4C).

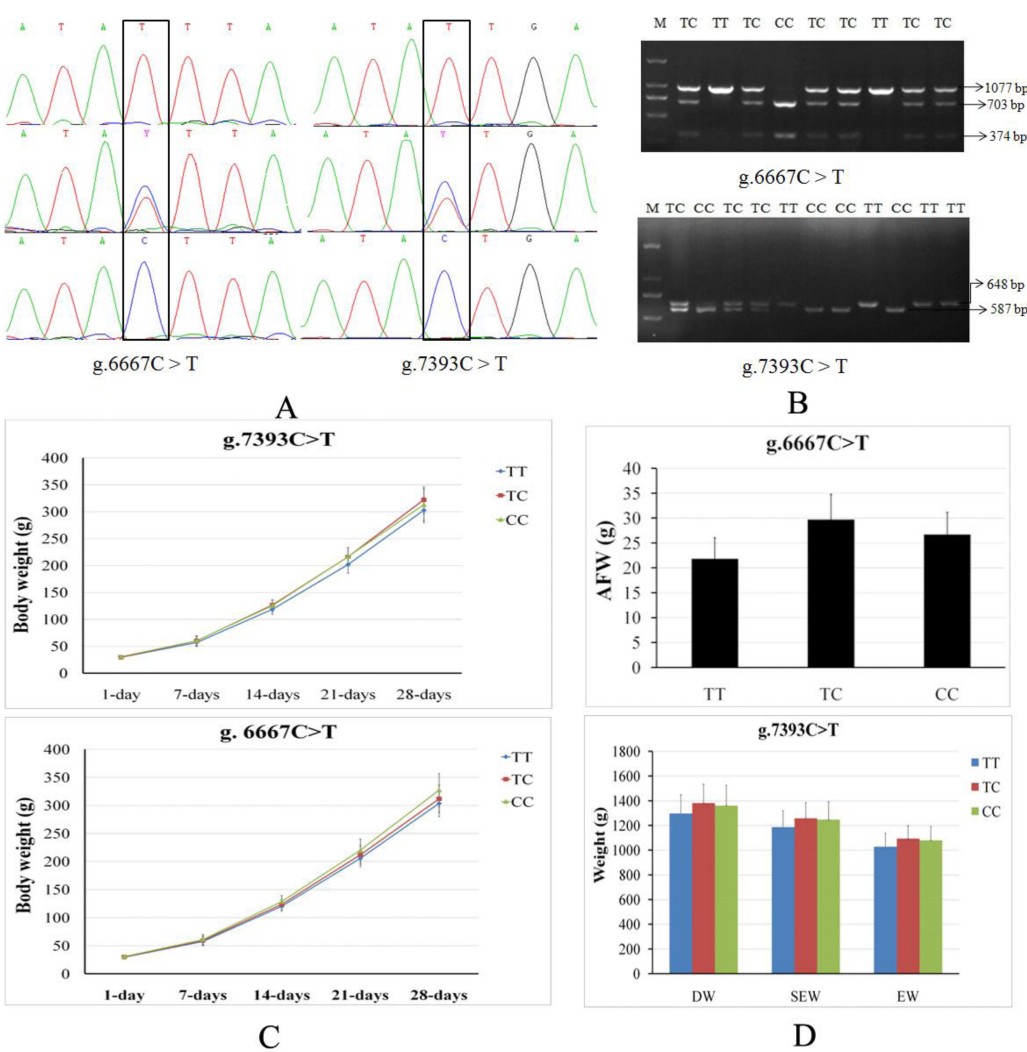

**Figure 2** **Two SNPs of cGNPDA2 gene associated with body weight and fatness traits in chicken.** (A) The sequencing profiles of the two SNPs in cGNPDA2 gene. R and Y are heterozygous, in which $R = A/G$ and $Y = C/T$. (B) Agarose gel results of two SNPs genotyped by PCR-RFLP. M, DNA marker; TT, TC and CC are three different genotypes. (C) The association of the two SNPs with the body weight of chickens during early stage of growth. (D) The association of the two SNPs with the fatness traits of chickens. DW, dressed weight; EW, eviscerated weight; SEW, semi-eviscerated weight.

The different *GNPDA2* genotypes may affect the gene expression levels; this could be a confounding effect on dietary status. Thus, we performed DNA sequencing to check the genotypes of all 45 chickens used for qPCR experiment (Fig. 5A). In 25 female chickens of 23 weeks old, the g.6667C > T have 5 CC, 12 TC and 8 TT genotypes; the g.7393C > T have 10 CC, 5 CT and 10 TT genotypes. We compared the *GNPDA2* expression level in these 25 chickens, the results showed that no significant difference were found ($P > 0.05$) among three different genotypes of both g.6667C > T and g.7393C > T (Fig. 5B and 5C).

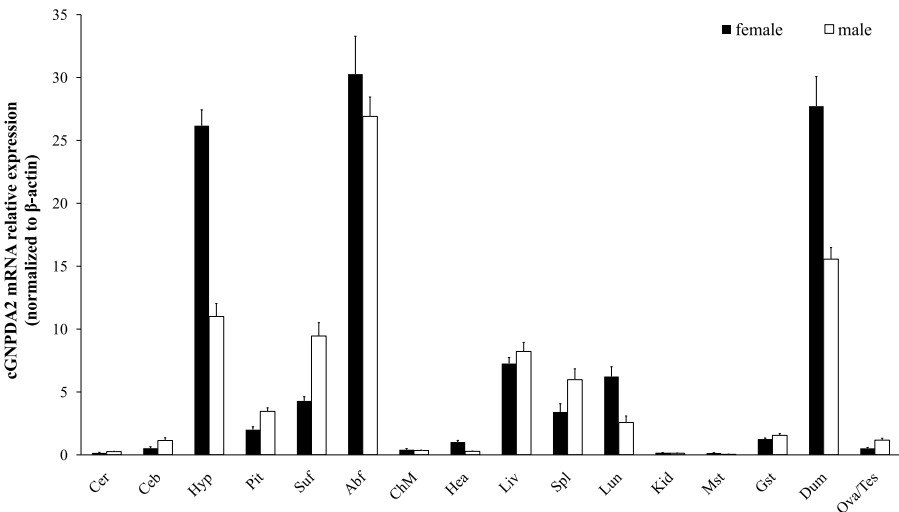

**Figure 3** **Expression of *GNPDA2* gene in adult XH chicken tissues.** The horizontal axis and vertical axis indicate different tissues and their mRNA relative expression values (mean ± S.E.M). Cer, cerebrum; Ceb, cerebellum; Hyp, hypothalamus; Pit, pituitary; Abf, abdominal fat; Suf, subcutaneous fat; Chm, breast muscle; Hea, heart; Liv, liver; Spl, spleen; Lun, lung; Kid, kidney; Mst, muscular stomach; Gst, glandular stomach; Dub, duodenum; Tes, testis; Ova, ovary.

**Table 2** **SNPs of *GNPDA2* gene associated with growth and carcass traits in chicken.**

| Markers | Traits | *P*-value | Least-squares mean ± S.E.M | | |
|---|---|---|---|---|---|
| g.6667C > T | CD (mm) | 0.0117 | 95.3 ± 9.15 (TT, 117) | 96.6 ± 9.26 (TC, 191) | 93.3 ± 9.32 (CC, 108) |
| | AFW (g) | 0.0006 | 21. 8 ± 4.23 (TT, 113) | 29.7 ± 5.05 (TC, 185) | 26.7 ± 4.46 (CC, 107) |
| | BW1 (g) | 0.0019 | 29.5 ± 2.40 (TT, 117) | 29.7 ± 2.81 (TC, 191) | 30.7 ± 2.74 (CC, 108) |
| | BW7 (g) | 0.0347 | 57.6 ± 7.63 (TT, 117) | 59.3 ± 8.50 (TC, 191) | 61.1 ± 9.12 (CC, 108) |
| | BW14 (g) | 0.0128 | 120.1 ± 8.68 (TT, 117) | 123.6 ± 9.61 (TC, 191) | 128.7 ± 10.53 (CC, 108) |
| | BW21 (g) | 0.0049 | 205.5 ± 14.80 (TT, 117) | 211.8 ± 16.47 (TC, 191) | 220.2 ± 19.81 (CC, 108) |
| | BW28 (g) | 0.0078 | 303.4 ± 23.35 (TT, 117) | 311.7 ± 24.41 (TC, 191) | 327.1 ± 29.58 (CC, 108) |
| g.7393C > T | DW (g) | 0.0114 | 1296.3 ± 153.9 (TT, 115) | 1381.2 ± 153.7 (TC, 130) | 1360.9 ± 164.7 (CC, 171) |
| | SEW (g) | 0.0026 | 1186.4 ± 133.8 (TT, 115) | 1257.7 ± 126.3 (TC, 130) | 1246.9 ± 144.7 (CC, 171) |
| | EW (g) | 0.0043 | 1027.6 ± 109.7 (TT, 115) | 1092.0 ± 105.5 (TC, 130) | 1079.0 ± 112.6 (CC, 171) |
| | LMW (g) | 0.0163 | 113.5 ± 12.6 (TT, 115) | 119.1 ± 12.5 (TC, 130) | 119.0 ± 14.9 (CC, 171) |
| | SIL (cm) | 0.0183 | 136.7 ± 11.4 (TT, 110) | 144.5 ± 13.4 (TC, 121) | 139.4 ± 10.8 (CC, 163) |
| | BW14 (g) | 0.0342 | 118.7 ± 9.06 (TT, 115) | 126.7 ± 10.49 (TC, 130) | 125.3 ± 9.40 (CC, 171) |
| | BW21 (g) | 0.0045 | 201.8 ± 15.99 (TT, 115) | 216.0 ± 17.61 (TC, 130) | 216.2 ± 17.29 (CC, 171) |
| | BW28 (g) | 0.0488 | 302.7 ± 23.42 (TT, 115) | 322.1 ± 24.84 (TC, 130) | 313.1 ± 27.23 (CC, 171) |

**Notes.**
AFW, abdominal fat pad weight; BW21, BW1, BW7, BW14, BW21 and BW28, body weight at 1,7,14, 21 and 28 days, respectively; CD, chest depth; DW, dressed weight; EW, eviscerated weight; LMW, leg muscle weight; SEW, semi-eviscerated weight; SIL, small intestine length.
Statistical analysis was performed using GLM in SAS software (version 9.0), and data are shown as mean ± S.E.M. Letters and numbers in bracket refer to genotype and number of chickens with that genotype.

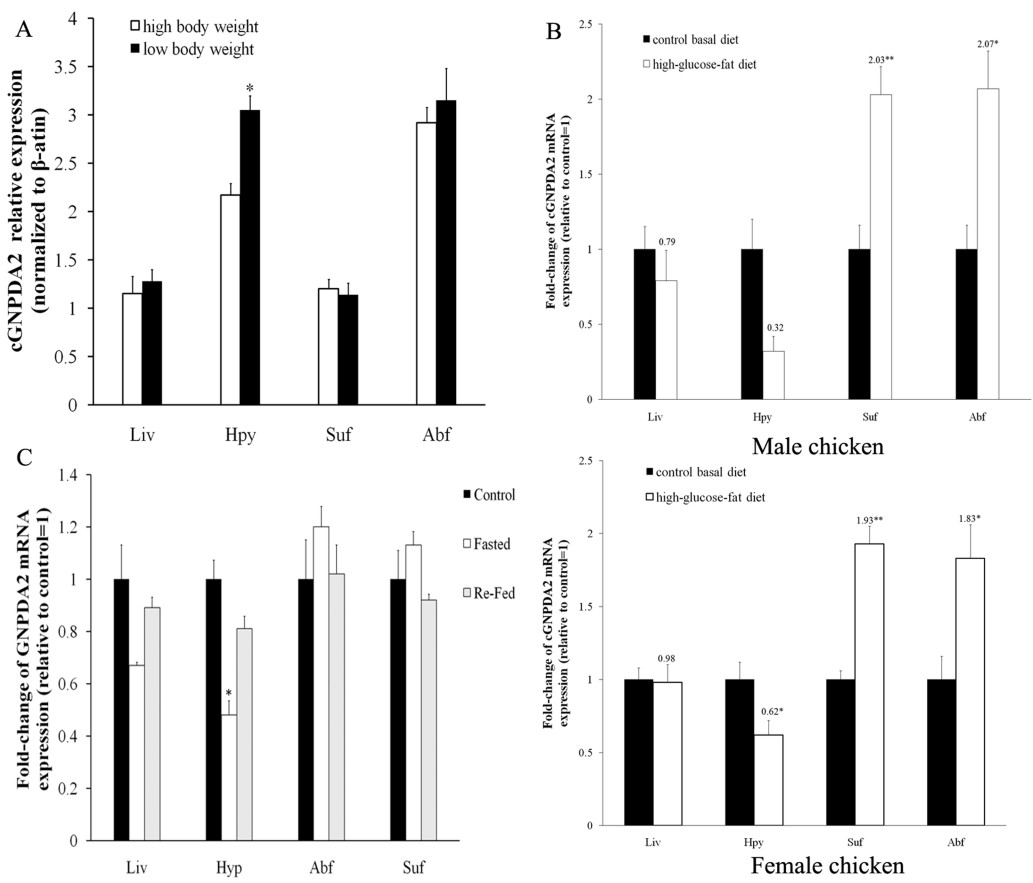

**Figure 4** **The level of c*GNPDA2* mRNA was regulated by body weight, feeding/fasting and by consumption of the high-glucose-fat diet.** (A) Level of c*GNPDA2* mRNA in XH chickens of high and low body weight. (B) Level of c*GNPDA2* mRNA in male and female XH chicken fed a basal diets and high-glucose-fat diets. (C) Changes in the level of c*GNPDA2* mRNA in XH chickens treated as control, fasted and Re-fed groups. Data are presented as the mean ± S.E.M ($n = 5$), *$P < 0.05$, **$P < 0.01$.

## Overexpression and knockdown of *cGNPDA2* in preadipocytes

In chicken preadipocytes, transfection of recombinant pcDNA3.1(+)-*GNPDA2*-a led to marked overexpression of *cGNPDA2* (7.32-fold higher, $P < 0.01$) and siRNA interfering also led to *cGNPDA2* decreased by 47% ($P < 0.05$). With overexpression of *cGNPDA2*, the mRNA level of *ACC* (acetyl-CoA carboxylase), *FAS* (fatty acid synthase), *FTO* (fat mass and obesity-associated gene) and *PGC1α* (peroxisome proliferative activated receptor gamma coactivator 1) were up-regulated, and the mRNA level of *ATGL* (adipose triglyceride lipase), *HL* (hepatic lipase), *PPARγ* (proxisome proliferator-activated receptor γ), *Leptin-R* (leptin receptor) and *PKM* (pyruvate kinase muscle) were down-regulated (Fig. 6A). When *cGNPDA2* was knocked down, the expression trend of related genes was just contrary to overexpression of *cGNPDA2* (Fig. 6B). Intriguingly, the mRNA level of *PKM* was decreased by 32% ($P < 0.05$) after overexpression *cGNPDA2*, and increased 1.76-fold ($P < 0.05$) after knockdown *cGNPDA2*. The expression of *PGC1α* was increased 2.25-fold ($P < 0.05$) after overexpression *cGNPDA2*. However, the changes of other related genes statistically were not significant difference after overexpression or knockdown of *cGNPDA2* ($P > 0.05$).

A

| Chicken groups for qPCR experiment | | Genotypes of g.6667C>T | | | | | Genotypes of g.7393C>T | | | | |
|---|---|---|---|---|---|---|---|---|---|---|---|
| | | 1 | 2 | 3 | 4 | 5 | 1 | 2 | 3 | 4 | 5 |
| 14 weeks old, male | Basal diet control | TC | CC | TT | CC | TT | CC | TC | CC | CC | TC |
| | High-glucose-fat diet | TC | CC | TC | TT | TC | CC | TT | CC | TC | TT |
| 14 weeks old, female | Basal diet control | CC | CC | TC | CC | TT | TC | TT | TT | TC | TC |
| | High-glucose-fat diet | TC | CC | TT | TT | TC | TT | TC | TT | CC | CC |
| 23 weeks old, female | High body weight | TC | CC | TC | CC | TT | CC | CC | TC | CC | TT |
| | Low body weight | TC | TC | TT | TC | TT | TC | TT | TT | CC | TT |
| | Fasted | TC | CC | TC | TC | TT | TT | TT | TT | CC | CC |
| | Re-fed | CC | TC | TT | TC | TT | CC | TT | CC | TT | TC |
| | Normal control | TT | TC | CC | TC | TT | TC | CC | TT | CC | TC |

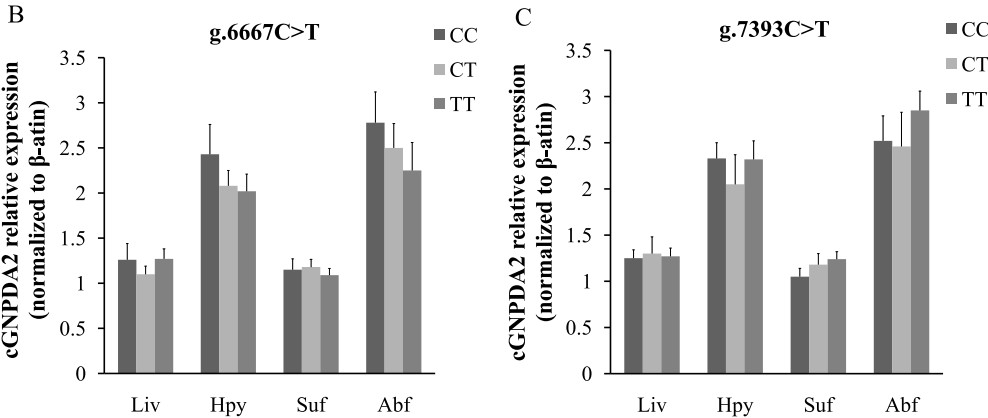

**Figure 5** **The different genotypes of g.6667C > T and g.7393C > T did not affected the expression of c*GNPDA2*.** (A) The genotypes of the chickens used for qPCR experiment. (B) The mRNA level of c*GNPDA2* of three different genotypes of g.6667C > T in 25 23-weeks-old female chickens. (C) The mRNA level of c*GNPDA2* of three different genotypes of g.7393C > T in 25 23-week-old female chickens.

## DISCUSSION

*GNPDA2* belongs to the GlcN6P subfamily, which encoded an allosteric enzyme of GlcN6P. Many *GNPDA2* transcripts were previously characterized in mammals, such as human, mouse and pig (*Leloir & Cardini*, *1956*; *Arreola et al.*, *2003*; *Li et al.*, *2010*). Four novel chicken *GNPDA2* transcripts (c*GNPDA2*-a∼c*GNPDA2*-d) were identified in this study. The predominant transcript *GNPDA2*-a has conserved domain of GlcN6P and share 85.5% amino acid identity with the other chicken GlcN6P (*GNPDA1*). This result is similar to that reported in human *GNPDA2*, which also share 87% amino acid identity with human *GNPDA1* (*Arreola et al.*, *2003*). Other chicken *GNPDA2* transcripts encode different short peptide, and did not contain the domain of GlcN6P. *GNPDA2* was reported to be conserved in mammalians (*Arreola et al.*, *2003*); here, we also used nucleotide BLAST analysis to detect homology between the amino acid sequence of *GNPDA2* among 18 species, and the result showed that *GNPDA2* was highly conserve among vertebrates, especially in birds.

The genomic structure of chicken *GNPDA2* is similar with human and mouse; they both comprised seven exons and six introns. The chicken *GNPDA2* gene spanned over 7 Kb in length and has many variations. Gene variations might have significant effects

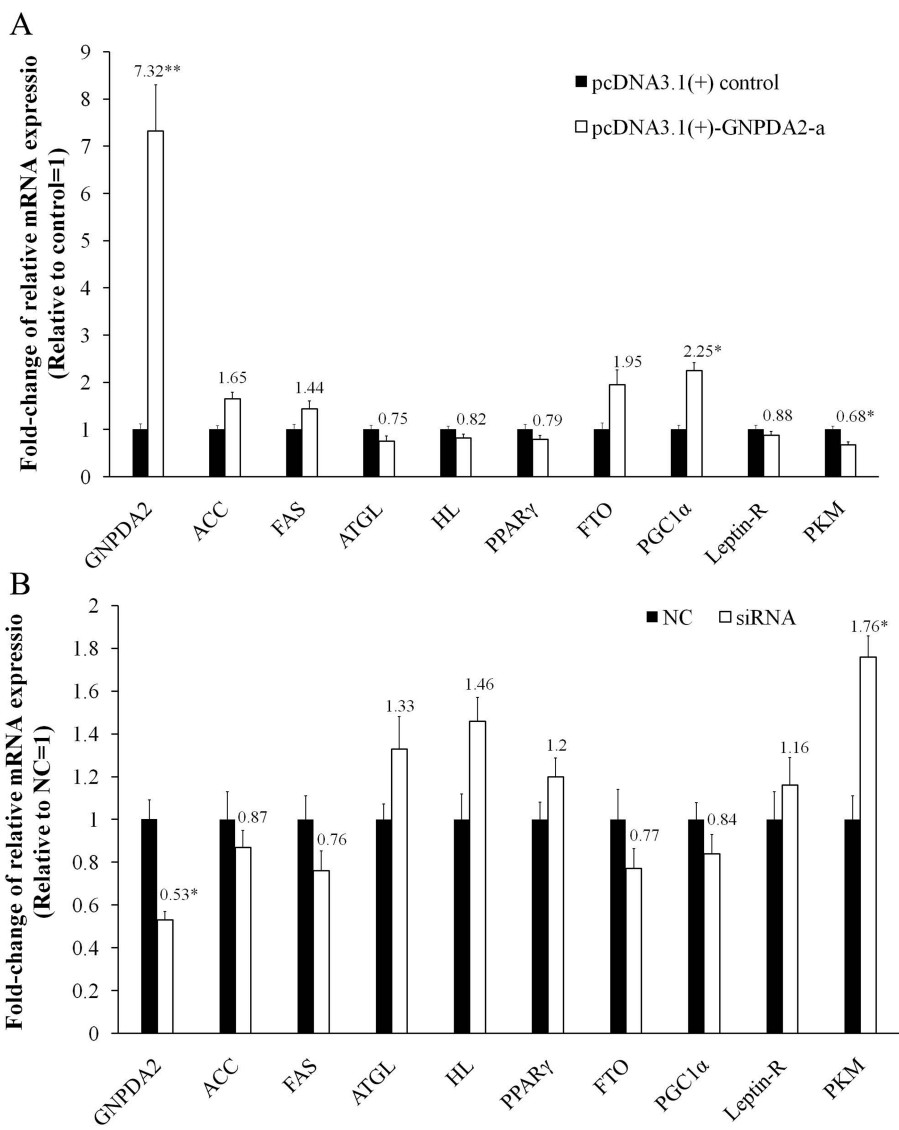

**Figure 6** **The changes of mRNA levels of related genes induced by cGNPDA2 overexpression (A) and siRNA interfering (B) in chicken preadipocytes.** Data are presented as the mean ± S.E.M ($n = 5$), *$P <$ 0.05, **$P < 0.01$.

on animal biology function. Previous studies showed that SNPs of human *GNPDA2* gene were associated with both BMI and body weight (*Willer et al.*, *2009*). Thus, we detected mutations of *cGNPDA2* and performed an association analysis with chicken production traits in our $F_2$ resource population. The results also showed that two of *cGNPDA2* SNPs were associated with chicken body weight and fatness traits.

The *GNPDA2* gene is expressed ubiquitously in mammalians (*Arreola et al.*, *2003*; *Skarnes et al.*, *2011*; *Graff et al.*, *2013*), and was found to be highly expressed in the hypothalamus of rats (*Schmid et al.*, *2012*). In the present study, we found the *GNPDA2* gene expressed in 17 various tissues of both male and female chicken. It also predominantly expressed in hypothalamus, duodenum and adipose tissues (subcutaneous fat and abdominal fat), and

the expression level were not significantly different between female and male chickens. *GNPDA2* gene plays an important role in hexosamine signaling pathway, which is one of the main nutrient-sensing pathways in organisms (*Marshall*, *2006*; *Oikari et al.*, *2016*). *GNPDA2* may be regulated in response to nutrition status or by dietary components. Previously study reported that the mRNA level of *GNPDA2* was down-regulated in the hypothalamus of rats fed with high fat diet (*Ruth et al.*, *2011*). In this study, we detected the change of *GNPDA2* expression with high-glucose-fat diet and fasted/refed conditions in hypothalamus, adipose tissue and liver when compared with controls (basal diet). The results showed that the expression of chicken *GNPDA2* gene was changed by different dietary status. Different genotypes of *GNPDA2* gene also may changes the gene expression. Therefore, we detected the genotype of these chickens and compared the *GNPDA2* expression level; the results showed that no significant difference were found among three different genotypes of both g.6667C > T and g.7393C > T. These results confirmed that the expression of *GNPDA2* was regulated by both feeding/fasting and by consumption of the high-glucose-fat diet and suggests a role of *GNPDA2* gene in fat and energy metabolism in chicken.

The regulation of fat metabolism, especially the function of *GNPDA2* in chicken preadipocytes (lipogenesis) is not yet clearly understood. We have found that variations of *cGNPDA2* were associated with chicken body weight and fatness traits. To further identify the functions of *GNPDA2*, we performed overexpression and siRNA interfering of *GNPDA2* in preadipocytes, and detected the change of fat and energy metabolism related genes. The results showed that the effects of *GNPDA2* on fat metabolism related gene were minor and are not significant ($P > 0.05$), such as fat synthesis gene (*ACC* and *FAS*), lipolysis gene (*ATGL* and *HL*), and preadipocytes differentiation gene (*PPARγ*). However, *PGC1α*, a key transcriptional coactivator of energy metabolism, was increased after overexpression of *GNPDA2* (*Espinoza & Patti*, *2005*; *Schuler et al.*, *2006*). *PMK* was down-regulated by *GNPDA2*, which plays a crucial role on glucose and energy metabolism (*Yang*, *2015*). These results suggest that *GNPDA2* might regulate energy metabolism through *PGC1α* and *PMK*, but the clear role of chicken *GNPDA2* in the regulation of fat and energy metabolism pathway still needs further study.

## CONCLUSIONS

In conclusion, herein we identified four novel chicken *GNPDA2* transcripts. The chicken *GNPDA2* gene was predominantly expressed in hypothalamus and adipose tissues. Its expression was regulated by both feeding/fasting and by consumption of the high-glucose-fat diet as compared with that of the controls. Overexpression of *GNPDA2* in chicken preadipocytes promoted the expression of *PGC1α* but inhibited *PKM*. Two SNPs of *cGNPDA2* gene were found to be markedly associated with body weight and fatness traits in chickens. Our data suggest that the *GNPDA2* gene is related to body weight, fat and energy metabolism in chickens.

## ACKNOWLEDGEMENTS

We thank the chicken farm workers of SCAU for their assistance in animal raising and also we would like to thank the two anonymous reviewers for their valuable comments and help for improving the manuscript.

### Funding

This work was funded by the National Natural Science Foundation of China (31172200) and the Program for New Century Excellent Talents in University (NCET-13-0803). The funders had no role in study design, data collection and analysis, decision to publish, or preparation of the manuscript.

### Grant Disclosures

The following grant information was disclosed by the authors:
National Natural Science Foundation: 31172200.
Program for New Century Excellent Talents in University: NCET-13-0803.

### Competing Interests

The authors declare there are no competing interests.

### Author Contributions

- Hongjia Ouyang conceived and designed the experiments, performed the experiments, analyzed the data, wrote the paper, prepared figures and/or tables, reviewed drafts of the paper.
- Huan Zhang performed the experiments, analyzed the data.
- Weimin Li performed the experiments, contributed reagents/materials/analysis tools, animal raising.
- Sisi Liang performed the experiments.
- Endashaw Jebessa performed the experiments, wrote the paper.
- Bahareldin A. Abdalla analyzed the data, reviewed drafts of the paper.
- Qinghua Nie conceived and designed the experiments, reviewed drafts of the paper.

### Animal Ethics

The following information was supplied relating to ethical approvals (i.e., approving body and any reference numbers):

Animal experiments were handled in compliance with the regulations and guidelines established by the Animal Care Committee of the South China Agricultural University (SCAU) (Guangzhou, People's Republic of China), and all efforts were made to minimize animal suffering. It was approved by the Animal Care Committee of SCAU with approval number SCAU#0011.

### DNA Deposition

The following information was supplied regarding the deposition of DNA sequences:
GenBank accession numbers: JX048609, JX048610, KF296359, KF296360.

## Data Availability

The research in this article did not generate any raw data.

## Supplemental Information

Supplemental information for this article can be found online at http://dx.doi.org/10.7717/peerj.2129#supplemental-information.

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
