# Peer review of "Identification, expression and variation of the GNPDA2 gene, and its association with body weight and fatness traits in chicken"

_PeerJ, doi:10.7717/peerj.2129_

## Round 0.1 · original submission · Major Revisions

Your paper has been reviewed by two experts. Both find it interesting and scientifically valid but agree that there is a need to improve the English writing and provide more details on the experimental protocols and main aims of the paper. I recommend that you have an expert in English review your paper before submitting a revision. Please consider carefully all the points made by the reviewers before submitting your revision.

Reviewer 1 ·

Basic reporting

This manuscript might be a better fit in a poultry-based journal.

There are extensive grammatical errors (inappropriate use of plurals or missing plurals, mixing up of prepositions, missing words, etc...).

Suggestion is to have an English-speaking colleague review this manuscript before next submission.

L30 productive = production? reproduction?
L62 no caps on metabolism and amino
L104 what is QG and QC, please define
L123,1L25 need to correctly cite programs and software throughout
L178 UCSC genome BLAT showed.. = UCSC genome BLAT was used to show...
L182 missed whole exon 5 = was deleted? was missing?
L190 Blasted analysis = Analysis by NCBI Blast (cite) showed...
there are others...

Experimental design

The authors could better define their research question, then tailor the discussion to provide evidence in support or not of their research question.
The basic kits used in the analysis are fairly standard. Methods appear straight forward except for the grammatical errors.
I couldn't find the titles or descriptions of the figures, so I can't comment on them.

Validity of the findings

The authors need to make a stronger case for why this is important to the poultry industry. This gene has been extensively evaluated in multiple species including humans. The authors set out "to characterize the functions of GNPDA2 gene in fat metabolism in chickens", but conclude that the genes they evaluated related to fat metabolism were not affected. This should be reflected in the structure of the discussion.

Additional comments

This manuscript might be a better fit in a poultry-based journal.

·

Basic reporting

The authors presented interesting data on the expression and SNPs of GNPDA2 gene in chicken fed on high fat diet or fasting. They found that the expression levels of GNPDA2 in different tissues are associated to the diet contents and that SNPs are associated to body weight. However, the quality of the data cannot really be estimated since a lot of precise information are missing or are not presented in a clear English. The most important one are the description in more detailed of the reverse transcription, the efficiency of each qPCR (see review of Bustin et al., 2009, Clinical Chemistry), the transfection details: amount of plasmid used for transfection, difference in cells viability, confluence, DNA preparation, the RFLP method…

Experimental design

As mentioned above the authors presented an effect of the different GNPDA2 genotypes on the body parameters and variation in expression levels of the GNPDA2 for similar traits, they should check the genotypes of the chicken used for qPCR experiment, because it could be a confounding effect (result Fig: 4 and 5).

Validity of the findings

This article suffers of pour English, and more detailed explanation and description of what was done.
The discussion should be more detailed with comparison to other results and discussion of the results.
However the findings are novel and interesting.

Additional comments

Fig 2 b is not really informative

---

## Round 0.2 · accepted · Accept

I did encounter a few minor grammatical errors. Presumably these can be fixed during the proof stage, or you may elect to send in a revised manuscript to fix the errors now. They are as follows:

abstract: the GNPDA2 gene have —> the GNPDA2 gene has

line 49: GNPDA2 gene encoded —> GNPDA2 gene encodes

line 134: 2.5 GNPDA2 sequences blast —> 2.5 GNPDA2 blast analysis

line 179: handled in compliance —> in compliance with what? Please specify.

Line 195: and phylogeny analysis —> and phylogenetic analysis

line 273: GNPDA2 is belongs to GlcN6P subfamily —> GNPDA2 belongs to the GlcN6P subfamily

line 293: GNPDA2 gene expressed —> The GNPDA2 gene is expressed

line 335: you may also want to thank the two anonymous reviewers for help improving the manuscript. Do you want to thank any funding organizations as well?